# Does the longevity of the Sardinian population date back to Roman times? A comprehensive review of the available evidence

Piergiorgio Floris[1], Maria Pina Dore[2,3], Giovanni Mario Pes [2,4]*

**1** Dipartimento di Lettere, Lingue e Beni culturali, University of Cagliari, Cagliari, Italy, **2** Dipartimento di Scienze Mediche, Chirurgiche e Sperimentali, University of Sassari, Sassari, Italy, **3** Baylor College of Medicine, Houston, Texas, United States of America, **4** Sardinia Longevity Blue Zone Observatory, Ogliastra, Italy

* gmpes@uniss.it

**Data Availability Statement:** All relevant data are within the manuscript and its Supporting Information files.

**Funding:** The authors received no specific funding for this work.

## Abstract

The discovery early in this century of the exceptional longevity of the Sardinian population has given new impetus to demographic studies of this phenomenon during the classical period. In the 1970s, it was hypothesised that the average mortality rate in Roman Sardinia was lower than in metropolitan Rome itself, postulating an ancient precedent for the remarkable longevity observable nowadays in the island's population. In the present study, the available evidence was examined in order to test this hypothesis. Literary, juridical, epigraphic, papyrological, anthropological and archaeological sources regarding the population of the Roman Empire, including Sardinia, were retrieved by accessing Science Direct, PubMed, Scopus and Google Scholar databases, as well as regional libraries, regardless of time limitation, and were independently reviewed by the authors. For Roman Sardinia, only funerary epitaphs were retrieved, in contrast with the numerous sources available for the whole Roman Empire. Inscriptions revealing the existence of three alleged nonagenarians, two centenarians, two ultracentenarians and one supercentenarian were found, corresponding to 2% in a total of 381 inscriptions. The majority were located in a highly Romanised rural area of central-western Sardinia. However, the ages reported in the epitaphs may be inaccurate because of the influence of confounders such as age rounding, approximations and/ or amplifications, and are unrelated to the total number of inhabitants. In conclusion, the funerary evidence, the only available data from Roman Sardinia, is too weak to estimate the life expectancy of the local ancient population and cannot offer valuable arguments to support the hypothesis that exceptional longevity has been a Sardinian trait since Roman times.

## Introduction

The twenty-year-old discovery of the exceptional longevity of people living on the Mediterranean island of Sardinia [1–4] has crowded the literature with studies focusing on the

**Competing interests:** The authors have declared that no competing interests exist.

demography of this ethnic group over a wide period of time [5–8]. This trend was somewhat anticipated by the interest in historical demographic research on ancient Sardinians. In the Roman age, that is, from the first century BCE to the third century CE, the publication of studies based on the interpretation of epigraphic data foreshadowed the development of a fruitful line of investigation [9–10]. Robert J. Rowland, Jr., in the 1970s, conjectured that the average rate of mortality in Roman Sardinia was surprisingly lower than that measurable in the population of metropolitan Rome in the same period [9]. Rowland's hypothesis, therefore, seemed to identify an ancient precedent for a demographic trait, such as longevity, currently observable in Sardinians.

Modern demographic research on Antiquity owes much to the analyses conducted by Karl Julius Beloch towards the end of the 19th century [11], and by Keith Hopkins since the 1960s [12, 13]. Current developments in this field have proceeded, roughly, since 1980 and have been animated by lively discussions on methodology and the use of sources [14–19]. Indeed, issues about the latter ones are still a subject of discussion among scholars. Walter Scheidel has pointed out that 'demography critically relies on numbers' [20]. Unfortunately, the main obstacle in the study of ancient demography lies precisely in the scarcity of data. In fact, there is a severe lack of information on all three components forming the basis of statistical analyses performed on modern and contemporary populations: birth rate, mortality and mobility of people [21]. Some authors have tried to investigate specific population subgroups whose age is sufficiently attested, such as that of the Roman emperors who died of natural causes, finding an unexpected concordance between their age at death and the information predicted by various statistical models [22].

The application to the Roman world of demographic models developed for human societies considered similar, closer in time and better known, has long been an important feature of the debate. In this regard, scholars tend to be divided between more or less sceptical (e.g. Walter Scheidel) and optimistic (e.g. Bruce Frier) approaches [16, 17]. The need to employ other types of sources, such as the scientific archaeology of human remains and the molecular biology of diseases, with the well-known advantages of using novel investigation methods, has been pointed out by Jongman [23].

On this basis, in this study we reviewed all available evidence about Roman Sardinia to answer the following questions: was mortality in Roman Sardinia comparable to that of other regions of the empire? Is it possible that the current longevity of the Sardinian population has its roots in Roman times?

## Materials and methods

### Search strategy and study selection

In order to accomplish the purpose of the study, a comprehensive search was conducted for any available literary, juridical, epigraphic, papyrological, anthropological or archaeological evidence by accessing multiple databases of the published literature, i.e. Science Direct (http://www.sciencedirect.com/), PubMed (http://www.ncbi.nlm.nih.gov/pubmed/), Scopus (http://www.scopus.com/home.uri) and Google Scholar (http://scholar.google.com/). All entries were examined, regardless of time or language limitation. Studies were identified using the following search query: 'Roman Empire', 'demography', 'mortality', 'Sardinia', 'Roman Sardinia', 'funerary inscriptions' or a combination thereof using the Boolean operators 'AND' and 'OR'. The review was conducted in accordance with the PRISMA guidelines as much as possible. In addition, for the literary and juridical sources we searched in the Digital Latin Library (DLL) (https://digitallatin.org/), in the Corpus Scriptorum Latinorum (CSL) (http://www.forumromanum.org/literature/), and in the Classical Latin Text (http://latin.packhum.org/).

For epigraphic and papyrological sources we retrieved data from the Epigraphic Database Roma (EDR) (http://www.edr-edr.it/default/index.php), and the Corpus Inscriptionum Latinarum (CIL) (http://cil.bbaw.de/), as well as the website Papyri Info (http://papyri.info/) which provides material from the Advanced Papyrological Information System (APIS), Duke Databank of Documentary Papyri (DDbDP), Heidelberger Gesamtverzeichnis der griechischen Papyrusurkunden Ägyptens (HGV), and Bibliographie Papyrologique (BP). Lastly, texts from the main Sardinian libraries, including the ones from the University of Cagliari and Sassari were retrieved and consulted. Publications containing descriptions of burial findings in Sardinia were independently and blinded examined by two authors (Floris P and Pes GM) and any resulting discrepancy was discussed among all authors to reach a final consensus. Documentary sources were considered eligible for the analysis on the basis of their reliability and similarity with the demographic structure of better known pre-modern and modern populations of Europe (18th century), and especially Asia (China), India and Egypt of the 19th and early 20th century [16, 20].

## Results and discussion

In the preliminary screening we were able to retrieve hundreds of articles, books and ancient inscriptions relating to Roman Sardinia. Articles quoting Sardinia marginally in the context of the Roman Empire, or without providing specific demographic data, were excluded from the analysis. Similarly, anthropological articles, reporting findings on skeletal remains in Sardinia were excluded if they did not encompass the Roman era, or if the age of the bones was not provided. Likewise, articles on Roman epigraphy that did not mention Sardinia were excluded. Following the preliminary screening only funerary epitaphs were retrieved for Roman Sardinia, including the texts of all 390 Latin funerary inscriptions reported by Rowland [9]. Moreover, articles of historical demography, from the end of the 19th century to the present, reporting data on the average lifespan in Sardinia compared to other regions of the Roman Empire were considered eligible and used to discuss similarities and differences.

### Literary sources

Ancient literary sources about mortality in Roman times are the outcome of an élite culture characterised by rhetorical and philosophical purposes [24]. Evidence concerning cases of exceptional longevity is not rare. However, the statistical significance of this information is invalidated by the absence of systematic method, a mixture of references to historical and mythical realities and the pervasive anecdotal and wondrous value that characterises this ancient material. Some of these features are found in a large section of Pliny the Elder's *Naturalis historia* (23–79 CE), which focuses on the topic of human life-span (Plin, Nat 7, 49, 153–164). An exemplary case is that of T. Fullonius, from Bologna, who in the census conducted by the emperor Claudius and L. Vitellius in 47–48 CE, declared himself to be 150 years old. Pliny reports that the emperor, intrigued, conducted cross-investigations on other documents relating to Fullonius, confirming the information's accuracy (Plin, Nat 7, 49, 159). After lingering on some astrological theories, relating to the maximum duration of human life (between 112 and 124 years), criticised by the author precisely for their variety (Plin, Nat 7, 49, 160–162) the antiquarian writer reported the existence of approximately 80 centenarians, ultracentenarians and supercentenarians who would have been identified in the VIII Augustan region of Italy (Aemilia) in the census of 73/74 CE. Pliny also reports that as many as 27 of them declared ages of between 110 and 140 years (Plin, Nat 7, 49, 162–164). Other ancient authors such as Valerius Maximus and Phlegon of Tralles underline with greater emphasis the 'prodigious' aspects found in the Plinian passage. The first, in his work *Factorum et dictorum*

*memorabilium libri IX*, written under Tiberius (14–37 CE), described persons who lived to an advanced age, proposing at the same time examples from reality and myth (Val Max. 9, 13). Phlegon, a freedman of the Emperor Hadrian (117–138 CE), in his *Opuscula de rebus mirabilibus et de longaevis*, reported a list of centenarians extracted from Roman census lists or literary tradition [25]. Notwithstanding the dubious credibility of the declarations made by these people to the census authorities, these data certainly attest to the presence of extremely old individuals in the Roman age, although they are not useful for statistical purposes.

## Juridical evidence

In the Justinian's Digest, there is a passage from the work '*Ad legem de vicesima hereditatum*' by the jurist Aemilius Macer (third century CE) that contains a comment concerning an Augustan law of 6 CE establishing a 5% tax on inheritance [26]. The passage mentions two schemes that attracted the attention of demography scholars: the first refers to the famous jurist Ulpian (c. 170–228 CE), while the second, albeit from an unknown epoch and author, is older. The schemes were used to estimate, on average and without distinction of sex, how long a life annuity could last in relation to the beneficiary's age, hence the definition of Ulpian's Life Table. Although it is not clear to which kind of people it refers (freeborn, ex-slaves, slaves, all) or how data were collected, it was thought that they–and mostly Ulpian's scheme–provide reference data to evaluate how long, on average, people of different ages could expect to live in the early third century CE [16, 27]. Among other inaccuracies, the table reveals an obvious gap in life expectancy estimation during early childhood, likely because children did not receive annuities [27]. However, Ulpian's Table seems to confirm the existence in Roman times of a dramatic infant mortality rate, a feature comparable to other pre-modern populations. Bruce Frier [16] proposed, in this regard, a comparison with Model West Level 2 of the Regional Model Life Tables of Princeton [28] and thus a low life expectancy at birth, around the age of 21 years. Despite the favourable consideration shown by Frier towards Ulpian's Table, most scholars are inclined to refuse its use to obtain information on the demographic structure of the Roman population [13, 14, 17]. Nonetheless, Robert Woods has recently found elements of contact between his novel demographic tables and Ulpian's Life Table [29].

## Epigraphic evidence

The tens of thousands of funerary inscriptions indicating age at death found in all areas of the Roman world have long been considered an important source for mortality and 'average' life expectancy. Many studies carried out in the past used data from funerary inscriptions to draw conclusions about the average age at death of people living in different regions of the Roman Empire. For instance, the Harkness's study, based on the inscriptions contained in the *Corpus of Latin Inscriptions*, reported the following average age at death in those who survived up to the age of 10 in various regions of the Roman Empire: Rome (29.3), Latium (29.6), Cisalpine Gaul (32.1), Bruttii, Lucania, Campania, Sicily and Sardinia (33.7), Calabria, Apulia and Samnium (34.8), England (36.5), Asia and Greece (36.8), Aemilia, Etruria and Umbria (37.1), Spain (37.8), and Africa (53.3). This early report highlighted the wide heterogeneity of mortality in the different parts of the Roman Empire and, specifically, the relatively longer life span in Sardinia compared with Rome itself [30].

The impressive catalogue of ages at death in the Roman era composed by János Szilágiy between 1961 and 1967 further highlights the interest in this material [31–36]. However, in 1966, Keith Hopkins claimed the inadequacy of funeral inscriptions as sources for demographic research: 'we cannot tell how much the longevity recorded in African inscriptions is the product of commemoration or of actual longevity' [12]. Epigraphic data would be

influenced, indeed, by uncontrollable socio-cultural and economic confounders [21], related to factors such as age, sex, location and living conditions. Therefore, with the partial exception of Bruce Frier who spared some North African epigraphic samples [16], most scholars consider funerary texts to be unreliable for studies on mortality rate and life expectancy [13, 14, 17, 20, 37–39]. Accordingly, studies carried out in the past on the ages retrieved from epitaphs – like Rowland's one about Sardinia [9] – did not provide information on population mortality and average life span, but only provided the median lifespan of the people commemorated in the texts [20].

Beyond the doubts regarding the demographic value of the epigraphic age indication, the attention of scholars focused also on their accuracy. First, it is well-known that, in Roman epitaphs, ages that are multiples of 5 are much more frequent than is admitted by statistics [40–42]. This observation led Tim Parkin to wonder whether the average Roman had an exact knowledge of his/her age, and whether at that epoch such a notion was as important to people as it is today [42]. The question was posed even though in many epitaphs biometric data are reported analytically with the indication of the number of years, months, days and even hours lived. In fact, Parkin formulated the hypothesis, perhaps worthy of further verification, that such scrupulous data may be related to the importance that the Romans attached to the celebration of their birthday and with their belief in astrology [42].

In conclusion, it seems better to consider funerary texts as the result of the commemorative practices of a specific area rather than as a faithful reflection of the demographic reality. Although these sources are useless for statistical analyses, their value remains high for social, cultural and economic investigations into commemoration mechanisms.

## Papyrological evidence

Papyrological source, which survived the particular climatic conditions of some areas of Egypt, are considered more reliable by Roman demography scholars. They consist of birth and death declarations included in about 300 acts related to provincial censuses, providing information on more than 1,100 people living in the first three centuries CE [16, 17]. Compared to other evidence of Roman provincial censuses, recently studied by Béatrice Le Teuff [43], those from Egypt have an intrinsic demographic value as they are more appropriate for mortality and life expectancy assessments than other evidence of Roman provincial censuses [16, 20]. In addition, Walter Scheidel considered papyri the cornerstone of mortality research [20]. The most exhaustive analyses of these documents have been performed by Roger Bagnall and Bruce Frier [44] and by Walter Scheidel [45], estimating life expectancy at birth in the order of 22.5 years, although it would be hasty to extend their results to the whole Roman Empire [17, 21]. Furthermore, the peculiar geographic and climatic conditions of Egypt, characterised by remarkable internal variability, contributed to the creation of distinctive pathogenic environments probably affecting the mortality and demographic structure of the local population [16, 17, 21].

## Archaeological evidence

Palaeodemographic investigations conducted on skeletal remains found in Roman burial sites aim at identifying sex, age at death and characteristics of the physical development of the deceased [46–49]. Skeletal anthropology also provided valuable information on eating habits, health status, diseases and causes of death [16], and in some cases it provided information on mortality aspects little represented in other types of sources, for example foetuses and children under 5 years of age [50]. However, Walter Scheidel noted at least two factors that make skeletal remains from a specific burial site not representative of local demography: not all members of a given population may have been buried in the same place due to socio-cultural

conditioning (age, gender, class); moreover, it seems impossible to measure the impact of migratory phenomena on the formation of the sample [17].

The excavations conducted in the necropolis of the Roman suburbs in recent decades have provided interesting anthropological and demographic data. The distribution of the ages at death shows that most of the deceased are in the age group of 20–49 years, and just over 4% refer to individuals over the age of 50 [51]. In addition, it was observed that most burials pertain to people belonging to lower social classes (slaves, freedmen) and several diseases affecting the population in the imperial age were identified. There are no data related to the age at death of skeletal remains in Sardinia of the Roman epoch, although physical anthropology research provided data for the prehistorical era in several sub-regions of Sardinia as described by Sanna et al. [52].

Methodologies to establish the demographic structure of ancient populations developed by palaeodemographic scholars are certainly promising and, in the future, they will play an important role in filling our knowledge gap.

### The comparative approach: Model Life Tables

The shortage of documentary sources, as well as uncertainty about their reliability, have led many scholars to adopt a comparative approach. Accordingly, scholars applied to the Roman Empire sets of demographic tables developed for more recent societies with a demographic structure assumed to be comparable to the Roman. In the 1960s, Keith Hopkins used the U.N. Model Life Tables [12] while afterwards, and up to the present day, the Regional Model Life Tables developed in Princeton since the 1960s by Ansley Coale and Paul Demeny were mostly adopted [13]. Even if specific Model Life Tables are sometimes used (e.g. Model South Level 3 Female, Model West Level 2, 4 or even 6), Model West Level 3 Female (adopted indifferently for men and women) is generally considered by scholars to be the most suitable to represent the Roman population [22].

The framework outlined by Model West Level 3 Female refers to a society characterised by remarkably high infant mortality and a life expectancy at birth of approximately 25 years. Infant mortality rates would have been balanced by equally high birth rates. Therefore, the Roman population would have been composed mostly of young people and adults up to the age of 50, while people over the age of 60 would have been much less numerous, and people over 80 rare [16, 42]. In a recent study, Robert Woods endorsed new sets of tables made in the 1970s, assuming a percentage of childhood deaths lower than usually expected, given the generally high levels of mortality. This would result in higher life expectancy at birth and higher mortality than previously thought for people aged between 15 and 49 years [29].

A specific approach to the use of demographic tables has been made by Bruce Frier since the beginning of 1980s, by using Model Life Tables to test data derived from ancient sources such as Ulpian's Life Table, skeletal remains found in some funerary sites and data from the Egyptian papyri [27, 44, 53]. Nevertheless, Frier observed that caution is required in the application of the tables to the Roman world, because the overall mortality of this vast area will have been the outcome of the demographic patterns of different regions of the Empire influenced by factors such as time, geography and perhaps individual social status [16]. Walter Scheidel excluded the notion that Model Life Tables can provide a reliable picture of Roman society, which was characterised by high mortality and considerable local variations in causes of death, considering ecology, climate and pathogenic environments as useful elements [21].

### Mortality in Roman Sardinia

The identification of the major causes of death in Roman society would be helpful to understand its demographic structure. Under normal conditions, or in the absence of acute events

such as the plagues of the second and third centuries CE [54, 55], the impact of infectious diseases on mortality was highly significant [17]. For example, the leading causes of death were malaria, diseases of the respiratory system (e.g. pneumonia and tuberculosis) and water- and food-borne epidemics (e.g. typhoid and cholera dysentery) [16]. However, the vastness and long duration of the Roman Empire make it improbable that identical causes acted everywhere and constantly in the same way. For example, differences existed between the pathogenic environments of the Alps and of the Egyptian marshy regions, and even over much shorter distances such as between rural and urban areas [21]. In the Mediterranean regions, the role of seasonal diseases such as malaria was undoubtedly relevant; in large geographic areas of Sardinia, fever caused by *Plasmodium falciparum* demanded a strong tribute of human lives [56], and associated malnutrition may have contributed to weakening the resistance of inhabitants, making them more susceptible to additional respiratory and gastrointestinal disorders [56–58]. Paleochristian funeral inscriptions have proved particularly useful in identifying the seasonal variation of deaths, as they often contain an indication of the day and month of death of the commemorated deceased [59–61]. In areas such as Sardinia, seasonality could be attributable to malaria, although not all Sardinian subregions were equally malarial, causing significant differences in local mortality and life expectancy patterns (Plin, Nat 7, 49, 162–164; Val Max. 9–13) [21]. Unfortunately, the prevalence and influence of malaria among adults in southern Italy including Sardinia remain unknown [62]. Moreover, modifications concerning climate, deforestation, urbanisation and extensive development of trade may have changed the epidemiological peculiarities of some areas over the centuries. On the other hand, these conditions could have been responsible for the spread of malaria or other diseases previously unknown in the West, such as those that caused the great epidemics of the second and third centuries CE [21, 63].

The reconstruction of the demography of Roman Sardinia suffers from all the difficulties described above for the whole Roman world, further worsened by an even lower availability of sources. First of all, the number and density of the island inhabitants is unknown. Karl Julius Beloch estimated that about 300,000 people lived in Roman Sardinia [11], later followed by Piero Meloni and Attilio Mastino [64, 65]. Meloni also suggested as a possible alternative half the number proposed by Beloch. However, both estimates are unverifiable, and it is unlikely that the number of islanders remained stable during the nearly seven centuries in which Sardinia was a Roman province. The demographic consequences of the brutal military operations conducted on the island by Romans in the third and second centuries BCE, the possible effects of the plagues of the second and third centuries CE and the impact of migration flows throughout the seven centuries of Roman rule, are totally unknown [65].

Mortality and average age were calculated in the past mostly on the basis of ages engraved in epitaphs. In the article '*Mortality in Roman Sardinia*' published in the 1970s, the North American scholar Robert J. Rowland Jr. reported the age at death of 390 Latin epitaphs (132 women and 238 men) dating back between the first and third centuries CE [9]. The weighted average age at death of Sardinians recorded on the inscriptions was 35.8 years (32.2 years for women and 37.6 years for men), i.e. much higher than the 23.4 years calculated for the city of Rome by using the same method. In a recent review of Sardinian data conducted on 381 valuable testimonials and contained in inscriptions datable between the first and third centuries CE [66] (Fig 1A), the number of epitaphs dedicated to children deceased between 0 and 9 years of age was overall 14%, rising up to 28% for young people who died between 10 and 19 years of age, while the percentage for people over 60 years was 18%, and about 4% and 1% for people aged over 80 and over 100, respectively. These results are largely in contrast with both the general framework described for a society with high mortality levels, and with the demographic tables themselves. Results from Sardinian epitaphs showed lower than expected age at

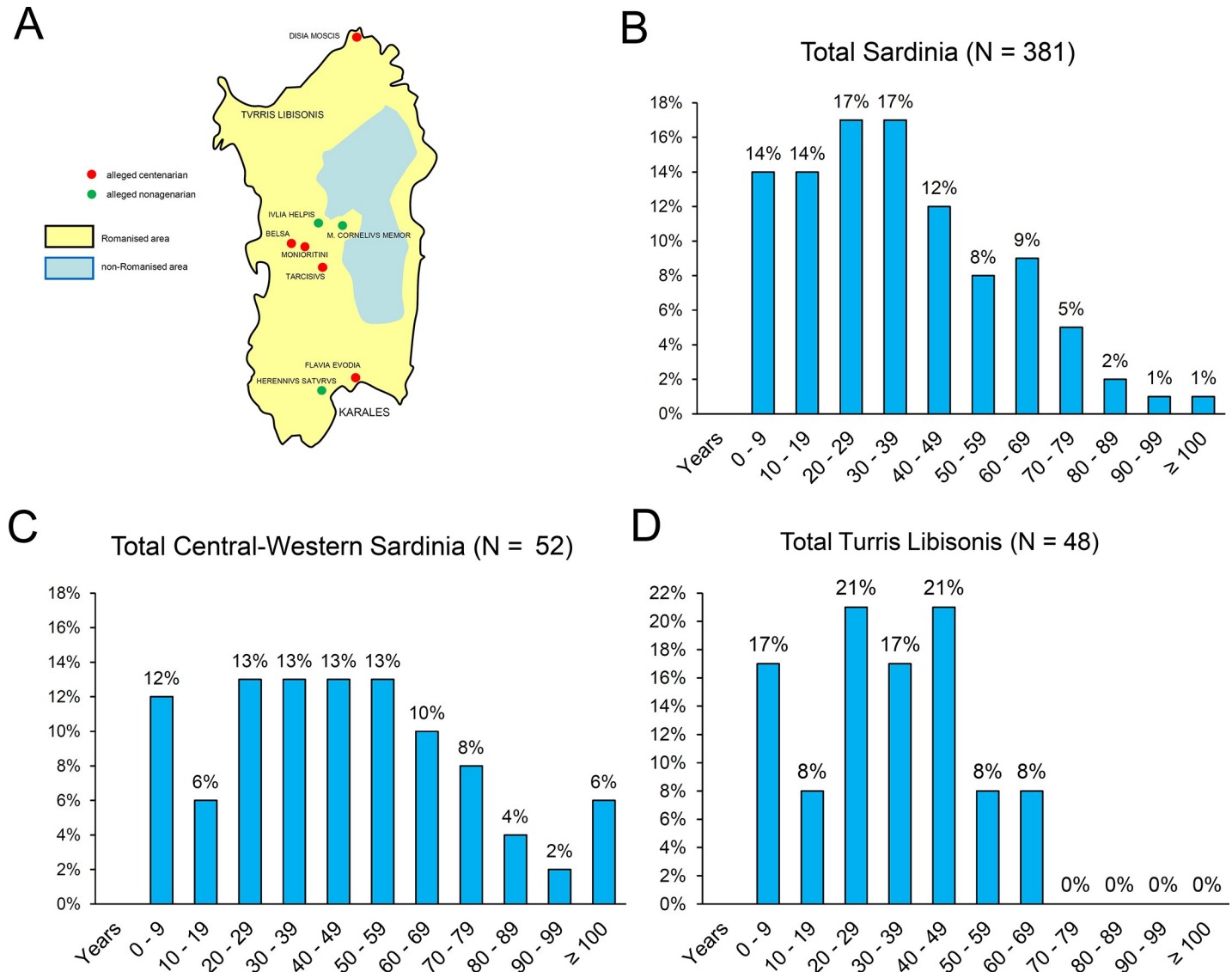

**Fig 1. Distribution of Sardinian epitaphs.** A. location of the epitaphs of alleged nonagenarians and centenarians in Roman Sardinia as well as of the two funerary clusters in northern (*Turris Libisonis*) and central-western areas of the island. B. Distribution of Sardinian epitaphs according to 10-year intervals; C. distribution of epitaphs in central-western Sardinia according to 10-year intervals; D. distribution of epitaphs in Turris Libisonis according to 10-year intervals.

death for children, and higher for the elderly, even though the longevity peaks of Roman Africa were not attained.

Epigraphic data can also be used to detect age differences across the island. Whereas in the rural areas of central-western Sardinia (Fig 1B) commemorations of the elderly reached the maximum levels (about 30% and 6% for people over the age of 60 and for individuals aged 100 or more years, respectively, out of nearly fifty valuable ages), in urban areas such as Turris Libisonis (= Porto Torres), the picture was completely different (Fig 1C). More specifically, about 8% of people were over 60 years of age out of almost fifty valuable ages, and no inscription was dedicated to people who died over the age of 69 years.

In the Roman funerary inscriptions of Sardinia dating between the first and third centuries CE, there is mention of three nonagenarians, two centenarians, two ultracentenarians and one supercentenarian (overall about 2% of the 381 testimonies screened). Five of them were found

in a region of central-western Sardinia with predominantly rural features. As far as we know, the oldest long-lived person was [T]arcisius son of Tarinci, a man who, according to his funerary inscription (discovered just before 1990 at *Filighes* in the territory of Ula Tirso) died around the first half of the second century CE at the age of 115 years (EDR142233 EDR = Epigraphic Database Roma, accessed at http://www.edr-edr.it/default/index.php) [67, 68]. The two ultracentenarians are 'Monioritini'—the name is almost certainly declined—and Belsa, died at 106 and 101 years of age, respectively. The epitaph of the former, perhaps dating to the first half of the first century CE, was also found in the Ula Tirso territory, in an area known as *Tilisai* [68], while Belsa's tomb was discovered in the plateau *Su Planu 'e pischinas*, a few kilometres away from the villages of Fordongianus and Paulilatino (EDR153075) [68]. In both cases, the names are likely indigenous; owing to this and to the lack of clear hints in the inscriptions, it is impossible to ascertain the gender of the deceased. Despite its more Roman or 'Romanised' sound, the name [T]arcisius could nonetheless have had a local origin [66, 68]. The indigenous background of the three centenarians is also presumable from the use (unquestionable for [T]arcisius and probable for the other two) of a particular type of funeral monument, the 'cippo a capanna' (hut-shaped tombstone). Nonetheless, the fact that their graves were accompanied by an inscription written in Latin suggests Roman cultural influence. These three testimonies from central-western Sardinia therefore seem to outline a quite precise picture: people from mainly rural areas imbued with the local indigenous background but influenced by Roman culture.

However, only three cases cannot have statistical relevance, and the alleged homogeneity of the picture does not seem to be confirmed by two other epigraphs from the same area mentioning nonagenarians. The first inscription, from *Sa Pala 'e sa Cresia* in the territory of Allai, recalls the ninety-one year old Iulia Helpis (EDR132678) [68, 69], while the second one, discovered in Austis, mentions the nonagenarian Ma(rcus) Cornelius Memor [68, 70]. From the onomastic point of view, both persons have little connection with [T]arcisius, Monioritini and Belsa. Despite the uncommon writing of his *prænomen*, Ma(rcus) Cornelius Memor has a fully Latin name, while Iulia Helpis, with her imperial *nomen gentilicium* and the Greek *cognomen*, represents almost an onomastic synthesis of the Graeco-Roman world. Even the social context of the two appears different: Ma(rcus) Cornelius Memor was perhaps an ex-soldier of the Roman garrison stationed in Austis [68, 71], while the status of Iulia Helpis, probably a former mistress or former slave of the man who commemorated her (the text does not allow a definitive conclusion), reflects a clear Roman cultural background.

Evidence of old people is not limited to the central-western part of Sardinia. Two other centenarians are known from inscriptions found in places located at the northern and southern ends of the island. A marble slab containing the epitaph of Disia Mosc(h)is (EDR081178) [72] was found in *Capo Testa* near Santa Teresa di Gallura in the mid-1980s. The woman, sister of the dedicator Eufrosinus (= Euphrosynus), would have died at 100 years. Although the age of Eufrosinus is unknown, as brother of the deceased he must have been old as well [72]. The onomastic of Disia Mosc(h)is and Eufrosinus has features very similar to those of Iulia Helpis and possibly shared the environment of freedmen and slaves.

The last testimony of a Sardinian centenarian was found in Karales (= Cagliari), the provincial capital, a populous city where the impact of malaria must have been far from negligible. During the 18th century, in the church of *San Lucifero* in Cagliari was found a *cupa* (barrel)—a type of funerary monument widespread in Karales [73] as well as in central-western Sardinia [74, 75]—bearing four epitaphs (EDR086433) [76]. Among the inscriptions, all concerning people from the same family, there is also one for Flavia Eu(h)odia, a woman dead at about (*plus minus*) 100 years, between the second half of the second and the first half of the third century CE. Indicators such as the imperial Flavian *nomen gentilicium*, the cognomen of Greek

origin and the use of a *cupa* suggest a servile past of the woman's family. The *plus minus* expression, referring to the years lived by Flavia Eu(h)odia, infrequent in inscriptions prior to the third century CE [76, 77], seems to indicate that her sons did not know the exact age of their mother. Finally, the epitaph of L(ucius) Herennius Saturus, who died at the age of 90 years, is engraved in a funerary altar found in the 1950s in Vallermosa, a town not far from Cagliari [78]. The name of the deceased appears to be fully Roman, although some scholars believe that the cognomen Satur(us), in areas heavily imbued with Punic traditions such as southern Sardinia, may represent the translation into Latin of a Punic personal name [66, 79].

The discovery of the exceptional longevity of the population of inner Sardinia at the turn of the 21$^{st}$ century raised the question of its historical origin and, in particular, of whether it could date back to the epoch of Roman rule. However, ascertaining the connection is difficult due to the scarcity of data relating to the Sardinian population in that period, as well as to the general uncertainty about the demography of the Roman population in Antiquity. In this review, we have examined the few documentary sources available in order to shed light on this problematic question.

Presumably, the Sardinian population in the Roman era overall exhibited characteristics similar to that of the rest of the empire, and comparable to other pre-industrial populations characterised by high mortality and high birth rates. In such conditions, life expectancy at birth likely ranged from approximately 20 to 30 years [12], though the possibility cannot be ruled out that both values, depending on local circumstances, were lower or higher [21]. Although the hypothesis that young people represented an important portion of the inhabitants of the Empire appears acceptable, the same cannot be affirmed for older people. Another factor that might have played a confounding role, and which must be taken into account in deciphering the demography of Sardinia in early centuries CE, is the different extension of Romanisation on the island, which has been documented for coastal areas and inland plains, but from which the central mountainous areas remained largely exempt due to the anti-Roman resistance of indigenous populations (Fig 1A).

Considering the available sources that have been reviewed, the literary, juridical, papyrological ones remain silent, and only the epigraphic and partly the archaeological ones offer a few glimpses about the existence of long-lived individuals in Sardinia. As far as the epigraphic sources are concerned, we reviewed the eight epitaphs spanning over three centuries (1$^{st}$-3$^{rd}$ CE) mentioning cases of extremely old persons. The ages of these long-livers may be inaccurate because of the influence of confounding factors, e.g. rounding, approximations, and/or amplifications, and are not related to the total number of inhabitants. Moreover, the study of these funerary inscriptions in Sardinia has shown a different distribution of the ages at death between the north (the Roman colony of Turris Libisonis) and the central western part of the island, which, apart from the occasional nature of funerary finds, seems to a large extent due to the different value attributed to the elderly in commemorative practices rather than to differences in the underlying demographic structure of the population. Importantly, the demographic weight of these alleged centenarians, ultra-centenarians and supercentenarians on the total population is unknown, because in the Roman world the access to 'memory' provided by the funeral inscription was a privilege of the few [76, 80]. These data hardly allow the conclusion of a large prevalence of elderly people of the island. The situation seems analogous to that of some regions of North Africa where epitaphs of old people are so frequent that they suggest the existence of a longevity hot-spot [50]. However, this hypothesis is now debated by scholars, in whose opinion African inscriptions have been altered by the local practices of commemoration, influenced by an extraordinary consideration of the elderly [12, 16, 45]. These explanations are largely plausible even though the question needs further investigation beyond the purposes of this paper.

## Conclusion

Our results indicate that, in the pagan funerary inscription of Roman Sardinia, there is not enough evidence to formulate definitive conclusions regarding people who lived beyond the age of 60. Although epitaphs can be of great importance for cultural history, in the absence of reliable demographic data, we cannot confirm that Roman Sardinia was 'a country of old men', nor that longevity in Sardinia is a mere extension of a phenomenon present since Roman times.

## Supporting information

**S1 Checklist. PRISMA checklist.**
(DOC)

## Author Contributions

**Conceptualization:** Piergiorgio Floris, Maria Pina Dore, Giovanni Mario Pes.

**Data curation:** Maria Pina Dore, Giovanni Mario Pes.

**Formal analysis:** Piergiorgio Floris, Maria Pina Dore.

**Funding acquisition:** Maria Pina Dore.

**Investigation:** Piergiorgio Floris, Maria Pina Dore.

**Methodology:** Maria Pina Dore.

**Project administration:** Maria Pina Dore.

**Resources:** Maria Pina Dore.

**Supervision:** Maria Pina Dore.

**Visualization:** Giovanni Mario Pes.

**Writing – original draft:** Piergiorgio Floris, Maria Pina Dore.

**Writing – review & editing:** Maria Pina Dore, Giovanni Mario Pes.

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
