## [Decision Letter · Decision Letter 0]

10 Sep 2020

PONE-D-20-23736

Does the Longevity of Sardinian Population Date Back to Roman Times? A Review of the Available Evidence

PLOS ONE

Dear Dr. Pes,

Thank you for submitting your manuscript to PLOS ONE. After careful consideration, we feel that it has merit but does not fully meet PLOS ONE’s publication criteria as it currently stands. Therefore, we invite you to submit a revised version of the manuscript that addresses the points raised during the review process.

We look forward to receiving your revised manuscript.

Kind regards,

David Caramelli, Ph.D

Academic Editor

PLOS ONE

Journal Requirements:

Reviewers' comments:

Reviewer's Responses to Questions

**Comments to the Author**

1. Is the manuscript technically sound, and do the data support the conclusions?

Reviewer #1: Partly

Reviewer #2: Yes

2. Has the statistical analysis been performed appropriately and rigorously? 

Reviewer #1: N/A

Reviewer #2: N/A

3. Have the authors made all data underlying the findings in their manuscript fully available?

Reviewer #1: Yes

Reviewer #2: Yes

4. Is the manuscript presented in an intelligible fashion and written in standard English?

Reviewer #1: Yes

Reviewer #2: No

5. Review Comments to the Author

Reviewer #1: This study reviews the literature in order to answer to the question if longevity of the Sardinian population dates back to Roman times.

The authors consider the characteristics and limits of different kinds of sources available for the Roman world (literary sources, juridical evidence, epigraphic evidence, papyrological evidence, model life tables, archeological evidence and causes of death) and then afford the analysis of data on mortality in Roman Sardinia.

My major concern is about the structure of the article, that does not correspond to the criteria for publication of the journal, as it is not a systematic review, nor it presents the results of original research.

However, the topic is potentially interesting for the readers of PLoS one.

My suggestion is to revise the article, slighthly changing the introduction on the evidence available for the Roman age (as suggested below) and realizing a systematic review of the literature on mortality in Roman Sardinia (according to PRISMA guidelines, http://prisma-statement.org/).

General comments

In the introduction, the authors should add a comparative, synthetic comment of the contribution given by the different sources to the knowledge of mortality in the Roman era. Otherwise, they risk to simply confirm in their conclusions (“Our results indicate that […] there is not enough evidence to formulate definitive conclusions ...”) what is already described in the introduction (“the usefulness of literary and epigraphic sources […] is today almost universally denied”, or “most scholars consider funerary texts unreliable for studies on mortality rate and life expectancy”).

I also suggest to introduce a comment on the utility of the information on longevity in the Roman age in order to better understand the environmental and biological correlates of longevity in the present Sardinian population. At this purpose, the discussion should include a comment on the diffusion of the Roman people in the different Sardinian subregions.

Furthermore, the anthropological literature should be considered with much greater attention.

Detailed comments

Abstract

> The abstract describes the background and the conclusions, without giving any information on the methods and the results of the study.

Introduction

> I suggest to remove “in the last quarter of the twenty century” as the information is already clearly present in the same or near sentences.

Mortality during the Roman empire

> The sentence “The scientific and methodological […] the use of sources“ is not very clear to me and should be reformulated.

Literary sources

>The use of multiple words to define concepts (e.g., rhetorical‒philosophical, technical‒statistical, anecdotal/exemplary/wondrous) is sligthly redundant.

> The comment “Even living aside […] scrutinized is unknown” could be better placed at the end of the chapter and joined to the other comment “It can be concluded that Roman age writers perceived longevity cases as extraordinary events and not as a daily reality”. However, the latter observation is not very informative.

Juridical evidence

> The statement “However, Ulpian's Table […] other pre-modern populations” is probably related to the following sentence. In such a case, please add the reference n.16.

Epigraphic evidence

> This chapter does not include demographic information, as done in the previous ones.

> Furthermore, I am quite surprised about the strength of the statement on the inadequacy of funeral inscriptions as sources for demographic research. Although the imprecision of age data undoubtedly hampers statistical analyses, such data can give qualitative information on the age structure of the population, that can be valuable in the absence of more accurate data. Indeed, age heaping is a phenomenon present in our days too (see for example studies on nutritional status in children not registered at birth), and it is considered in order to manage and save the related information.

Papyrological evidence

> If possible, please add some information on mortality or longevity, as in the first paragraphs.

Archeological evidence

> This chapter does not consider the great body of information on demographic, biological and pathological characteristics furnished by the anthropological literature.

> A comment on the imprecision of age of death assessment would be appropriate.

The comparative approach: Model Life Tables

> Please discuss why the Model West Level 3 Female tables (Table 1) are “the most often quoted in the literature” and if this is true also for studies on longevity, where men and women generally show a different trend (with the exception of eastern Sardinia).

Causes of death

> Here, again, the anthropological literature is underscored. Just as an example, consider the article by Minozzi et al. “Palaeopathology of Human Remains from the Roman Imperial Age” (https://doi.org/10.1159/000338097). Some other information, specifically referred to Sardinia, can be retrieved in the book authored by Sanna (Il popolamento della Sardegna e l'origine dei sardi; CUEC, 2006) and in the online archive http://www.anthroponet.it/.

> The conclusions (“in conclusion […] beyond the purposes of this paper”) should be placed under a dedicated chapter summarizing the relevance of the different sources in a comparative way. I also suggest to add a table with the key information deriving from or related to the sources previously discussed.

Mortality in Roman Sardinia

> In this section, the authors refer to a recent review of one of them, mentioned as forthcoming within the references (Floris, P. Forthcoming: Anziani, centenari e ultracentenari nella documentazione epigrafica della Sardegna romana) and it is unclear if the figure 1 is original or comes from that article. The information on the graphic elaboration included in the legend can be given in the appropriate section on authors' contribution.

> The authors should justify why, in their opinion, “epigraphic evidence suggests different trends in the commemorative practices […] than in Turris Libisonis. “

> The sentence “it must be remembered that […], the funerary altar […] was found in the 1950s” should be changed. In fact, it seems that the important fact to be remembered is the year of the altar discovery.

Conclusions

> These conclusions are only related to the last chapter and not consider the first part of the article, that are summarized before. The previous conclusions should be moved here or the title changed.

Figures 2-4. Just a remind that these figures will be shared under an open access licence.

References

> References should be written according to the “Vancouver” style used by PLOS. Actually some references are footnotes.

Reviewer #2: Manuscript "Does the Longevity of Sardinian Population Date Back to Roman Times? A Review of the Available Evidence" in the first part (maybe a little too long compared with the rest) describes the source and the limits of method for estimating the age of death in Roman period, then it focuses the attention on Sardinia Case.

Authors made an important review work, but in my opinion the manuscript needs major revision before publishing.

The most important point is the structure of the manuscript. I suggest to re-organize the manuscript following the structure indicated by Plos One: introduction, materials and methods, results, discussion, and eventually Conclusion. Any other paragraphs should be included in the previous cited as subparagraphs.

Second page:

- Financial Disclosure: The are no findings for this study. Maybe authors mean There is no funding.

- Re-organize the abstract, highlighting the aim of the paper and the type of research (review)

Introduction: In my opinion would be useful to write in the introduction all the possible source or statistical estimation for the age of death, while in material and method author should cite the source they consulted for the bibliography (database and keywords, library), since the manuscript is a review of the available literature, we must be sure to have all the available literature.

In Introduction:

- When it is possible, it would appreciate to specify, when authors cite Romans, if is know which part of Roman Empire they are speaking about;

- When authors write that there is a trend to interpret the longevity observed nowadays in Sardinia as a historical extension of intrinsic characteristics of this population documentable in the Roman era, they should cite the source, who gave this interpretation?

The last part of “Causes of death” does not fit well in that paragraph, I suggest to move it in the previous one.

In “mortality of Roman Sardinian”:

- when mortality rates for malaria is cited, author should specify the value;

- Author cite the age of death of Sardinia compared with the one of the city of Rome, are there other data from other part of Roman Empire?

- The sentence: “only three cases are not statistically very significant” is scientifically wrong: a statistic can be significant, highly significant or nor significant, other adjectives are not accepted. Authors could write for example 3 cases not have statistical relevance or similar;

- Only study based on epitaph are mentioned, I presume no other source are available. Authors should specify it.

English must be revised by a native speaker experienced in the field.

Figures: Despite being fascinating, in my opinion figures do not add value or information to the manuscript. Maybe authors could limit the number of figures.

6. PLOS authors have the option to publish the peer review history of their article (what does this mean?). If published, this will include your full peer review and any attached files.

Reviewer #1: No

Reviewer #2: No

---

## [Author Response · Author response to Decision Letter 0]

28 Nov 2020

Authors’s responses to Reviewers

Reviewer #1: This study reviews the literature in order to answer to the question if longevity of the Sardinian population dates back to Roman times.

The authors consider the characteristics and limits of different kinds of sources available for the Roman world (literary sources, juridical evidence, epigraphic evidence, papyrological evidence, model life tables, archeological evidence and causes of death) and then afford the analysis of data on mortality in Roman Sardinia.

My major concern is about the structure of the article, that does not correspond to the criteria for publication of the journal, as it is not a systematic review, nor it presents the results of original research.

Reply: the structure of the article was completely reshaped according to the Plos ONE criteria and the title was changed as follows “Does the longevity of the Sardinian population date back to Roman times? A comprehensive review of the available evidence?”

However, the topic is potentially interesting for the readers of PLoS one.

My suggestion is to revise the article, slighthly changing the introduction on the evidence available for the Roman age (as suggested below) and realizing a systematic review of the literature on mortality in Roman Sardinia (according to PRISMA guidelines, http://prisma-statement.org/).

Reply: we have done so.

General comments

In the introduction, the authors should add a comparative, synthetic comment of the contribution given by the different sources to the knowledge of mortality in the Roman era. Otherwise, they risk to simply confirm in their conclusions (“Our results indicate that […] there is not enough evidence to formulate definitive conclusions ...”) what is already described in the introduction (“the usefulness of literary and epigraphic sources […] is today almost universally denied”, or “most scholars consider funerary texts unreliable for studies on mortality rate and life expectancy”).

I also suggest to introduce a comment on the utility of the information on longevity in the Roman age in order to better understand the environmental and biological correlates of longevity in the present Sardinian population. At this purpose, the discussion should include a comment on the diffusion of the Roman people in the different Sardinian subregions.

Furthermore, the anthropological literature should be considered with much greater attention.

Reply: in the revised version of the manuscript, we moved the second paragraph to the introduction, and added a supplement of the literature on mortality in the Roman Empire that we consider exhaustive (Page 4, lines 74-77; page 5, lines 81-84). In the introduction, the diversity of opinions of various authors on the duration of life in Roman times was emphasized, thus avoiding a predictable statement in the conclusions (Page 4, lines 66-77). In particular, the prevalent negative opinion of scholars about the reliability of epigraphic, funerary sources, etc. for the estimate of mortality was not anticipated in the introduction but developed from the critical analysis of evidence. The discussion includes a comment on the different extension of Romanization in the various subregions of Sardinia (Page 19, lines 407-415).

Detailed comments

Abstract

> The abstract describes the background and the conclusions, without giving any information on the methods and the results of the study.

Reply: methods and results were added to the abstract, that was deeply changed.

Introduction

> I suggest to remove “in the last quarter of the twenty century” as the information is already clearly present in the same or near sentences.

Reply: we did it.

Mortality during the Roman empire

> The sentence “The scientific and methodological […] the use of sources“ is not very clear to me and should be reformulated.

Reply: the sentence was simplified as follows: “Modern demographic research on Antiquity owes much to the analyses conducted by Karl Julius Beloch towards the end of the 19th century [11], and by Keith Hopkins since the 1960s [12, 13]. Current developments in this field go on, roughly, since 1980 and are animated by lively discussions on methodology and the use of sources [14-19].” Page 4, lines 66-69.

Literary sources

>The use of multiple words to define concepts (e.g., rhetorical‒philosophical, technical‒statistical, anecdotal/exemplary/wondrous) is sligthly redundant.

Reply: the use of multiple words is a common expression in the literary field and may refer to complementary aspects of the ancient world. In the revised version we tried to avoid the combination of these terms using the symbol “/” by inserting a conjunction. The word “technical / statistical” was eliminated. See, for instance, page 6, lines 109 and 112.

> The comment “Even living aside […] scrutinized is unknown” could be better placed at the end of the chapter and joined to the other comment “It can be concluded that Roman age writers perceived longevity cases as extraordinary events and not as a daily reality”. However, the latter observation is not very informative.

Reply: in the revised manuscript the last sentence was suppressed.

Juridical evidence

> The statement “However, Ulpian's Table […] other pre-modern populations” is probably related to the following sentence. In such a case, please add the reference n.16.

Reply: we did it.

Epigraphic evidence

> This chapter does not include demographic information, as done in the previous ones.

Reply: in the revised manuscript we added a paragraph on page 8, lines 161-166.

> Furthermore, I am quite surprised about the strength of the statement on the inadequacy of funeral inscriptions as sources for demographic research. Although the imprecision of age data undoubtedly hampers statistical analyses, such data can give qualitative information on the age structure of the population, that can be valuable in the absence of more accurate data. Indeed, age heaping is a phenomenon present in our days too (see for example studies on nutritional status in children not registered at birth), and it is considered in order to manage and save the related information.

Reply: in the revised manuscript the sentence was suppressed, although the majority of scholars in the field deny the utility of epigraphic data for demographics purposes. 

Papyrological evidence

> If possible, please add some information on mortality or longevity, as in the first paragraphs.

Reply: thanks for the kind observation. Data on life expectancy were added to the papyrological section (Page 10, line 205).

Archeological evidence

> This chapter does not consider the great body of information on demographic, biological and pathological characteristics furnished by the anthropological literature.

> A comment on the imprecision of age of death assessment would be appropriate.

Reply: the archaeological evidence section is now more informative and some comments were added (Page 10, lines 222-223; page 11, lines 224-232).

The comparative approach: Model Life Tables

> Please discuss why the Model West Level 3 Female tables (Table 1) are “the most often quoted in the literature” and if this is true also for studies on longevity, where men and women generally show a different trend (with the exception of eastern Sardinia).

Reply: the main reason for the utility of Model West Level 3 Female is that it fits empirical data better than any other. In the literature this model was adopted irrespectively of sex because in pre-modern societies such as ancient Romans the survival of men and women was similar. A sizable gender gap in survival is appreciable only in populations characterised by some evidence of “longevity”, e.g. survive beyond the age of 60. For these reasons the Model West Level 3 Female cannot be applied to study longevity.

Causes of death

> Here, again, the anthropological literature is underscored. Just as an example, consider the article by Minozzi et al. “Palaeopathology of Human Remains from the Roman Imperial Age” (https://doi.org/10.1159/000338097). Some other information, specifically referred to Sardinia, can be retrieved in the book authored by Sanna (Il popolamento della Sardegna e l'origine dei sardi; CUEC, 2006) and in the online archive http://www.anthroponet.it/.

> The conclusions (“in conclusion […] beyond the purposes of this paper”) should be placed under a dedicated chapter summarizing the relevance of the different sources in a comparative way. I also suggest to add a table with the key information deriving from or related to the sources previously discussed.

Reply: information on the article by Minozzi et al. and the data for the book of Sanna were included in the Mortality in Roman Sardinia section (Page 11, lines 224-229) and, in general, the anthropological literature was expanded. As regard the table, in Sardinia there are only epigraphic documentation making no possible to compare the relevance of the different sources (Page 11, lines 227-229).

Mortality in Roman Sardinia

> In this section, the authors refer to a recent review of one of them, mentioned as forthcoming within the references (Floris, P. Anziani, centenari e ultracentenari nella documentazione epigrafica della Sardegna romana, in press) and it is unclear if the figure 1 is original or comes from that article. The information on the graphic elaboration included in the legend can be given in the appropriate section on authors' contribution.

Reply: in the revised manuscript, figures 2-4 were suppressed. Only figure 1 was retained.

> The authors should justify why, in their opinion, “epigraphic evidence suggests different trends in the commemorative practices […] than in Turris Libisonis.”

Reply: as for the different commemorative practices, the answer can be found in the specific section of the revised manuscript (Page 19, lines 407-415)

> The sentence “it must be remembered that […], the funerary altar […] was found in the 1950s” should be changed. In fact, it seems that the important fact to be remembered is the year of the altar discovery.

Reply: in the revised version the sentence was removed.

Conclusions

> These conclusions are only related to the last chapter and not consider the first part of the article, that are summarized before. The previous conclusions should be moved here or the title changed.

Reply: we did it.

Figures 2-4. Just a remind that these figures will be shared under an open access licence.

Reply: see above (figures 2-4 have been suppressed).

References

> References should be written according to the “Vancouver” style used by PLOS. Actually some references are footnotes.

Reply: in the revised manuscript some quotations from ancient writers have been moved from the reference list directly into the text, within parentheses. In addition, references have been modified according to the Vancouver style.

Reviewer #2: Manuscript "Does the Longevity of the Sardinian Population Date Back to Roman Times? A Review of the Available Evidence" in the first part (maybe a little too long compared with the rest) describes the source and the limits of method for estimating the age of death in Roman period, then it focuses the attention on Sardinia Case.

Authors made an important review work, but in my opinion the manuscript needs major revision before publishing.

The most important point is the structure of the manuscript. I suggest to re-organize the manuscript following the structure indicated by Plos One: introduction, materials and methods, results, discussion, and eventually Conclusion. Any other paragraphs should be included in the previous cited as subparagraphs.

Reply: In the revised version, the structure of the manuscript was completely changed according to the instruction of Plos ONE, although the historical nature of the article does not completely fit into a conventional format.

Second page:

- Financial Disclosure: The are no findings for this study. Maybe authors mean There is no funding.

- Re-organize the abstract, highlighting the aim of the paper and the type of research (review)

Reply: The misspelling relating to funding has been corrected. In the abstract it was clearly stated that the paper is a review.

Introduction: In my opinion would be useful to write in the introduction all the possible source or statistical estimation for the age of death, while in material and method author should cite the source they consulted for the bibliography (database and keywords, library), since the manuscript is a review of the available literature, we must be sure to have all the available literature.

In Introduction:

- When it is possible, it would appreciate to specify, when authors cite Romans, if is know which part of Roman Empire they are speaking about;

- When authors write that there is a trend to interpret the longevity observed nowadays in Sardinia as a historical extension of intrinsic characteristics of this population documentable in the Roman era, they should cite the source, who gave this interpretation?

Reply: a detailed methodology for source retrieval has been described in the material and methods section (Page 5, lines 91-100; page 6, lines 101-104).

When available, regions of the Roman Empire are specified.

In the present work now is better specified that we tested the hypothesis that the extreme longevity detected in some areas of Sardinia was a phenomenon already present in Roman times, as suggested by the Rowland’s report (Page 4, lines 60-65).

The last part of “Causes of death” does not fit well in that paragraph, I suggest to move it in the previous one.

Reply: we did it.

In “mortality of Roman Sardinian”:

- when mortality rates for malaria is cited, author should specify the value;

Reply: No reliable death rate from malaria is available for Roman Italy as reported in the added reference no. 62.

- Author cite the age of death of Sardinia compared with the one of the city of Rome, are there other data from other part of Roman Empire?

Reply: the average age at death of Sardinia in comparison with other parts of the Empire was included (Page 8, lines 161-168).

- The sentence: “only three cases are not statistically very significant” is scientifically wrong: a statistic can be significant, highly significant or nor significant, other adjectives are not accepted. Authors could write for example 3 cases not have statistical relevance or similar;

Reply: The sentence on statistical significance was changed according to the reviewer's suggestion.

- Only study based on epitaph are mentioned, I presume no other source are available. Authors should specify it.

Reply: this was specified on page 2, line 28-29; page 18, lines 396-398; page 19, lines 421-424.

English must be revised by a native speaker experienced in the field.

Reply: a certification by an English proof-reading service was attached as Supporting Information files.

Figures: Despite being fascinating, in my opinion figures do not add value or information to the manuscript. Maybe authors could limit the number of figures.

Reply: as suggested by the reviewer, figures 2-4 have been suppressed.

---

## [Decision Letter · Decision Letter 1]

14 Dec 2020

PONE-D-20-23736R1

Does the longevity of the Sardinian population date back to Roman times? A comprehensive review of the available evidence?”

PLOS ONE

Dear Dr. Pes,

Thank you for submitting your manuscript to PLOS ONE. After careful consideration, we feel that it has merit but does not fully meet PLOS ONE’s publication criteria as it currently stands. Therefore, we invite you to submit a revised version of the manuscript that addresses the points raised during the review process.

We look forward to receiving your revised manuscript.

Kind regards,

David Caramelli, Ph.D

Academic Editor

PLOS ONE

Reviewers' comments:

Reviewer's Responses to Questions

**Comments to the Author**

1. If the authors have adequately addressed your comments raised in a previous round of review and you feel that this manuscript is now acceptable for publication, you may indicate that here to bypass the “Comments to the Author” section, enter your conflict of interest statement in the “Confidential to Editor” section, and submit your "Accept" recommendation.

Reviewer #1: All comments have been addressed

Reviewer #2: All comments have been addressed

2. Is the manuscript technically sound, and do the data support the conclusions?

Reviewer #1: Yes

Reviewer #2: Yes

3. Has the statistical analysis been performed appropriately and rigorously? 

Reviewer #1: N/A

Reviewer #2: N/A

4. Have the authors made all data underlying the findings in their manuscript fully available?

Reviewer #1: No

Reviewer #2: Yes

5. Is the manuscript presented in an intelligible fashion and written in standard English?

Reviewer #1: Yes

Reviewer #2: Yes

6. Review Comments to the Author

Reviewer #1: I have read the article with renewed interest and believe that it is significantly improved.

The authors have revised the manuscript according to suggestions. However, revision is still necessary, as detailed below.

I understand the difficulty to completely adapt the manuscript to the PRISMA guidelines. However, something more can be done.

Within the section Materials and methods (Search strategy and study selection) more information should be given (e.g., when the bibliographic search was performed, the years considered, the search query). Within the Results, if not a classical flow diagram, the numbers of studies screened, assessed for eligibility, and included in the review, with reasons for exclusions should be added. Also, a brief description of the characteristics of selected studies is necessary. For example, the information included in the abstract “For Roman Sardinia, only funerary epitaphs were retrieved” should result evident from this section. Indeed, it is unclear if the studies discussed in pages 6-18 have been selected for the review or if they are mentioned for comparison. Hence, it is unclear if the Results are overlapped with the Discussion.

Detailed comments

Lines 63-65. This sentence “On this basis […] of this population” is related to the aim ot the review and should be joined with it (lines 85-87).

Line 179. I do not see the reason to write median under quotes. Please explain.

In general. The authors use many terms when refer to the analysis of skeletal remains (scientific archaeology, molecular biology, paleodemography), but strangely never “anthropology” (or skeletal anthropology, or physical anthropology) were the study of such topic is central.

Lines 407-415. According to the authors' answer, this text “includes a comment on the different extension of Romanization in the various subregions of Sardinia”. However I dont see where.

Fig 1. Distribution of Sardinian epitaphs. Please add in the legend and possibly in the figure the information on the northern location of Turris Lisbonis. The same at line 319.

Reviewer #2: The authors addressed all the reviews I suggested and the manuscript is now greatly improved, so it should be considerated accept.

I just sugget to modify some little points more:

- In Introduction: line 70, I suggest to eliminate recently, because a research published in 2007 is not recently in my opinion;

- In Introduction: line 81: the authors name Frier, never mentioned before, I suggest to add a reference here;

- In Discussione, line 393: authors wrote "Although he hypothesis". It is not clear who is he, please substitute with the name.

In Discussione line 408: please add a comma after because, or eliminate the comma after world in line 409

7. PLOS authors have the option to publish the peer review history of their article (what does this mean?). If published, this will include your full peer review and any attached files.

Reviewer #1: No

Reviewer #2: No

---

## [Author Response · Author response to Decision Letter 1]

18 Dec 2020

Authors’s responses to Reviewers

Reviewer #1: I have read the article with renewed interest and believe that it is significantly improved.

The authors have revised the manuscript according to suggestions. However, revision is still necessary, as detailed below.

I understand the difficulty to completely adapt the manuscript to the PRISMA guidelines. However, something more can be done.

Within the section Materials and methods (Search strategy and study selection) more information should be given (e.g., when the bibliographic search was performed, the years considered, the search query). Within the Results, if not a classical flow diagram, the numbers of studies screened, assessed for eligibility, and included in the review, with reasons for exclusions should be added. Also, a brief description of the characteristics of selected studies is necessary. For example, the information included in the abstract “For Roman Sardinia, only funerary epitaphs were retrieved” should result evident from this section. Indeed, it is unclear if the studies discussed in pages 6-18 have been selected for the review or if they are mentioned for comparison. Hence, it is unclear if the Results are overlapped with the Discussion.

Reply: in the revised version of the manuscript the section on materials and methods has been enriched with new information to facilitate the reader in understanding the search strategy. In particular, the following paragraph has been added “In the preliminary screening we were able to retrieve hundreds of articles, books and ancient inscriptions relating to Roman Sardinia. Articles quoting Sardinia marginally in the context of the Roman Empire, or not providing specific demographic data, were excluded from the analysis. Similarly, anthropological articles, reporting findings on skeletal remains in Sardinia were excluded if they did not encompass the Roman era, or the age of the bones were not provided. Likewise, articles on Roman epigraphy that did not mention Sardinia were excluded. Following the preliminary screening only funerary epitaphs were retrieved for Roman Sardinia, including the texts of all 390 Latin funerary inscriptions reported by Rowland [9]. Moreover, articles of historical demography, from the end of the 19th century to the present, reporting data on the average lifespan in Sardinia compared to other regions of the Roman Empire were considered eligible and used to discuss similarities and differences.” (page 6, lines 116-119; page 6, lines 120-126).

On page 5, line 94 it was specified that the search had no time limitation.

Although some articles retrieved from the search did not specifically deal with Sardinia, they were nevertheless cited in the text to clarify some peculiar aspects of demography in Roman times.

According with the manuscript structure of PLOS One Journal, Results and Discussion was combined in one section (page 6, line 115).

Detailed comments

Lines 63-65. This sentence “On this basis […] of this population” is related to the aim ot the review and should be joined with it (lines 85-87).

Reply: Thanks for the suggestion. The sentence has been shortened and moved to the last paragraph of the introduction (page 5, line 82).

Line 179. I do not see the reason to write median under quotes. Please explain.

Reply: Sorry for the mistake, in the revised version the quotes have been suppressed (page 10, line 200).

In general. The authors use many terms when refer to the analysis of skeletal remains (scientific archaeology, molecular biology, paleodemography), but strangely never “anthropology” (or skeletal anthropology, or physical anthropology) were the study of such topic is central.

Reply: as suggested by the reviewer, in the revised manuscript the term “anthropology” has been added accordingly, and now there are 5 occurrences of the word, included one in the abstract (page 2, line 24; page 5, line 90; page 11, line 235; page 12, lines 244 and 249).

Lines 407-415. According to the authors' answer, this text “includes a comment on the different extension of Romanization in the various subregions of Sardinia”. However I dont see where.

Reply: A new sentence was added: “A factor that might have played a confounding role, and which must be taken into account in deciphering the demography of Sardinia in early centuries CE, is also the different extension of the Romanization on the island, clearly documented for coastal areas and inland plains, while the central mountainous areas remained largely exempt due to the anti-Roman resistance of indigenous populations (Figure 1 A)” (page 19, lines 415-419).

Fig 1. Distribution of Sardinian epitaphs. Please add in the legend and possibly in the figure the information on the northern location of Turris Lisbonis. The same at line 319.

Reply: a new figure 1 has been prepared, in which a stylized geographical map has been added displaying (i) the extension of Romanized Sardinia compared to areas relatively less influenced by the Roman rule; (ii) the location of Turris Libisonis in the North of 'island, and (iii) the site of the funerary findings relating to alleged ultra-centenarians and nonagenarians.

Reviewer #2: The authors addressed all the reviews I suggested and the manuscript is now greatly improved, so it should be considerated accept.

Reply: We appreciate your feedback and evaluation very much.

I just sugget to modify some little points more:

- In Introduction: line 70, I suggest to eliminate recently, because a research published in 2007 is not recently in my opinion;

Reply: We agree with the reviewer, and in the revised manuscript the adverb “recently” has been deleted.

- In Introduction: line 81: the authors name Frier, never mentioned before, I suggest to add a reference here;

Reply: The reference to Frier has been quoted (page 5, line 78).

- In Discussione, line 393: authors wrote "Although he hypothesis". It is not clear who is he, please substitute with the name.

Reply: Sorry for the typo about "he hypothesis"; in the revised version it has been corrected with “the hypothesis” (page 19, line 413).

In Discussione line 408: please add a comma after because, or eliminate the comma after world in line 409

Reply: the comma was moved (page 20, line 432).

We hope the revised manuscript will be now suitable for publication in the PLOS One Journal 

Sincerely and respectfully

Giovanni Mario Pes, MD, PhD

Dipartimento di Scienze Mediche, Chirurgiche e Sperimentali

University of Sassari

Viale San Pietro no. 43

I-07100 Sassari

Italy

Phone: +39 347 4539532

E-mail: gmpes@uniss.it

---

## [Decision Letter · Decision Letter 2]

21 Dec 2020

Does the longevity of the Sardinian population date back to Roman times? A comprehensive review of the available evidence?”

PONE-D-20-23736R2

Dear Dr. Pes,

We’re pleased to inform you that your manuscript has been judged scientifically suitable for publication and will be formally accepted for publication once it meets all outstanding technical requirements.

Kind regards,

David Caramelli, Ph.D

Academic Editor

PLOS ONE

Additional Editor Comments (optional):

Reviewers' comments:

Reviewer's Responses to Questions

**Comments to the Author**

1. If the authors have adequately addressed your comments raised in a previous round of review and you feel that this manuscript is now acceptable for publication, you may indicate that here to bypass the “Comments to the Author” section, enter your conflict of interest statement in the “Confidential to Editor” section, and submit your "Accept" recommendation.

Reviewer #1: All comments have been addressed

2. Is the manuscript technically sound, and do the data support the conclusions?

Reviewer #1: Yes

3. Has the statistical analysis been performed appropriately and rigorously? 

Reviewer #1: N/A

4. Have the authors made all data underlying the findings in their manuscript fully available?

Reviewer #1: Yes

5. Is the manuscript presented in an intelligible fashion and written in standard English?

Reviewer #1: Yes

6. Review Comments to the Author

Reviewer #1: (No Response)

7. PLOS authors have the option to publish the peer review history of their article (what does this mean?). If published, this will include your full peer review and any attached files.

Reviewer #1: No

---

## [Editor Report · Acceptance letter]

26 Dec 2020

PONE-D-20-23736R2 

Does the longevity of the Sardinian population date back to Roman times? A comprehensive review of the available evidence 

Dear Dr. Pes:

I'm pleased to inform you that your manuscript has been deemed suitable for publication in PLOS ONE. Congratulations! Your manuscript is now with our production department. 

Kind regards, 

on behalf of

Professor David Caramelli 

Academic Editor

PLOS ONE